# Multistep Evolution Method to Generate Topological Interlocking Assemblies

**Andres Bejarano** *  and **Kathryn Moran**

Department of Computer Science, Purdue University, West Lafayette, IN 47907, USA; klmoran@purdue.edu
* Correspondence: abejara@purdue.edu

**Abstract:** Research on topological interlocking (TI) assemblies indicates that the geometry of blocks plays a significant role in the performance of a configuration. The current TI generation methods can return assemblies of uniform antiprisms, tetrahedra, cubes, and octahedra. However, other shapes (both convex and concave) are well qualified for use in TI assemblies. This paper presents a framework to generate blocks for TI assembly. Starting from a seed polygon, evolution steps translate and reshape the polygon, contracting it eventually to a point, a line segment, or another polygon. Our framework generalizes and unifies previous-generation methods based on tilting angles and height parameters. We show how the proposed method systematically generates novel TI solids and previously reported others.

**Keywords:** topological interlocking; generation; mid-section; evolution steps

## 1. Introduction

The topological interlocking (TI) principle states that two adjacent blocks interlock if their respective top and bottom evolution sections (cross-sections) degenerate into a line segment or a point. Dyskin et al. defined this principle [1,2], and Kanel-Belov et al. later formalized it [3]. The Platonic solids and uniform antiprisms satisfy the TI principle since we can find polygons that describe their mid-sections with respective evolutions as cross-sections.

There are relevant instances of TI assemblies throughout architectural history. Fallacara discussed the case of topological variation in the flat vault (examples in Figure 1) and its relevance in history [4,5]. In the author's words, "*The historical problem is to find a building solution to cover a space with a flat floor constituted by discrete elements: in other words, to build a vault with zero radius stone-ashlar*" [4]. Fallacara discussed the flat vault designs from Joseph Abeille (in 1699) and Jean (Sébastien) Truchet (in 1704), both considered examples of topological interlocking assemblies centuries before such a term was coined [6].

To generate a TI assembly, we start with a surface tessellation composed of convex tiles with an even number of sides each. A generation method designs an interlocking block per tile. The traditional generation method [3] requires angle parameters that describe the tilted faces of the blocks. A recently proposed method named Height–Bisection [7] introduces height parameters to generate valid, block-aligned TI assemblies even for non-planar surface tessellations. Figure 2 shows the generation methods for generating an octahedron using either the traditional method or the Height–Bisection method. However, both methods can only generate uniform antiprisms and three of the Platonic solids: the tetrahedron, the cube, and the octahedron. We require the post-processing of the blocks to generate other known TI block shapes or clip overlapping, misaligned blocks. For example, to generate a truncated tetrahedron, we need to generate a traditional tetrahedron and then truncate its vertices adequately. Designers and engineers must modify the blocks either manually or computationally.

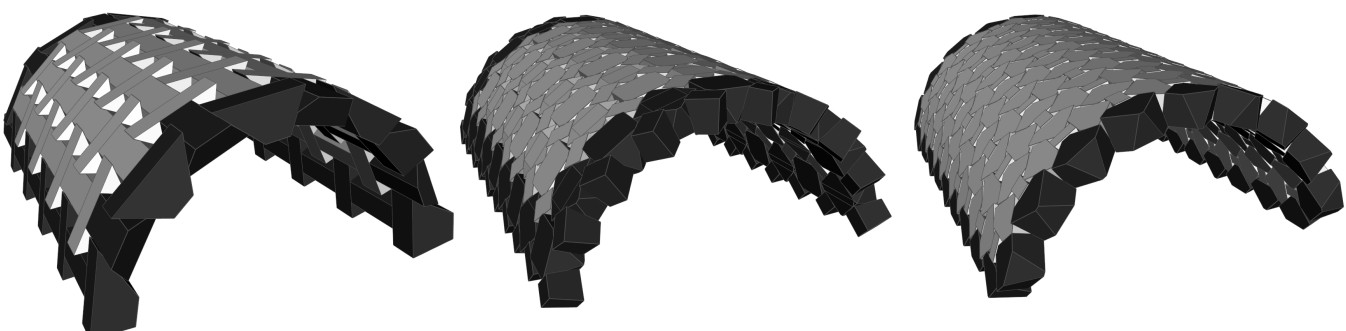

**Figure 1.** Examples of flat vaults designed as topological interlocking assemblies made of convex blocks. (**Left**): truncated tetrahedra. (**Middle**): truncated square antiprisms and truncated tetrahedra. (**Right**): truncated octahedra and truncated tetrahedra.

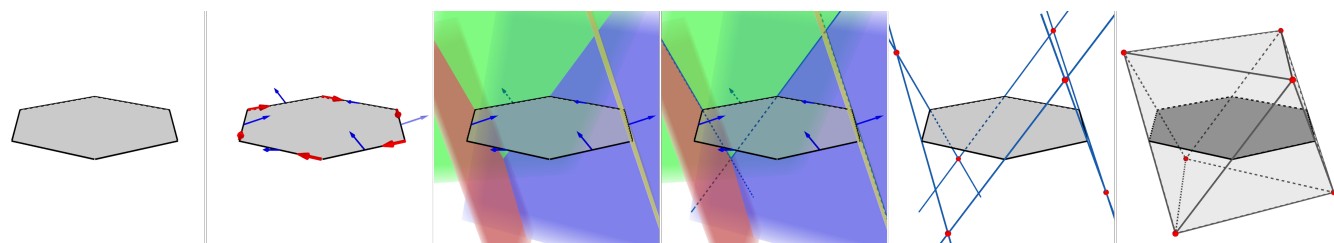

**Figure 2.** Generation steps for an octahedral TI block. From **left** to **right**: tessellation tile, edge parameters, tilted incident planes to edges (backplanes shown for visualization), intersection lines between planes, intersections between lines, and interlocking blocks.

Recent work on the performance of TI assemblies shows that block shapes play a significant role in the integrity of a structure. Mirkhalaf et al. considered the shapes of the interlocking pieces in their study of the strength and toughness of TI materials [8,9]. Multiple assembly designs (based on square and hexagon tiles) were 3D-printed and tested. The authors found that increasing the tilting angles of the contact surfaces improved the energy absorption of the TI. Weizmann et al. considered the relation between the geometry of the blocks and the structural performance of the assembly [10]. The authors considered a variety of planar TI assemblies based on different interlocking geometries. Every configuration was subjected to indentation tests that measured the displacement of the pieces after applying loads. Their results showed that a higher number of tiles and block faces with lower tilting angles reduced the structure's strength. Finally, Williams and Siegmund described how the geometry of the blocks affected the paths traversing a TI assembly under loads [11].

In this article, we report a generalization of the mid-section evolution concept. It allows for the design of any block with TI properties. We approach the concept by including evolution steps for a given polygon such that its evolution along a direction vector produces the corresponding block vertices. Each step requires a collection of angle and distance parameters that reshape and translate a polygon from one evolution step to another. The evolution parameters are similar, in nature and purpose, to the parameters of the Tilting Angles and Height–Bisection methods.

We begin with a discussion of the current work on the TI principle, focusing on the geometry of the blocks. Next, we describe the polygon evolution process, which passes through several evolution steps. Finally, we introduce the General Mid-Section evolution concept, which includes a generation method and the fundamental TI generation requirement. We conclude with examples of TI solids generated, including the Platonic, truncated, and concave solids with TI properties.

## 2. Related Work

Many polyhedra comply with the TI principle. The list includes Platonic solids, certain Archimedean solids, and uniform antiprisms. Additionally, some concave shapes (primarily based on convex counterparts) also comply with the interlocking principle by having face-to-face contact only (i.e., no joinery or connectors) in an assembly.

### 2.1. Block-Shape Analysis

The traditional TI shapes are the Platonic solids. Dyskin et al. reported that the Platonic solids can maintain TI when assembled in planar sections [2]. Each solid results from the evolution of specific tile shapes. A square evolves into a tetrahedron. A hexagon evolves into an octahedron, cube, or dodecahedron. A decagon evolves into a dodecahedron or an icosahedron. A surface tessellation of decagons requires additional kite-like tiles to fill the gaps. In such a case, the generation method must ignore the kite-like tiles to avoid unexpected results. The truncated versions of the Platonic solids also have interlocking properties. Dyskin et al. argued that "*certain truncations of [the Platonic] solids leave the interlocking property unaffected*" [2]. Glickman [12] and Dyskin et al. [13] give examples of interlocking assemblies made of truncated tetrahedra. Furthermore, Dyskin et al. [14] described three different arrangements for an assembly of buckyballs (i.e., truncated icosahedra) to be interlocking (associated with the symmetries of fifth and third orders).

Recent advances concerning geometrical approaches come from computational architecture. Weizmann et al. used TI assemblies to build facades, prioritizing their work on extending the catalog of resultant convex interlocking assemblies based on different types of tessellations [15]. Their conclusion is a connection between the geometry of the blocks and the structural performance of the assembly. Their work in [16] continued the search for an expanded catalog of TI assemblies. They concluded that three semi-regular tessellations are appropriate for TI purposes: $3.12^2$, $4.6.12$, and $4.8^2$. Furthermore, the authors state that there is no limit for TI assemblies based on non-regular tessellations since their number is infinite. The authors also considered the support structure required to assemble floors made of convex interlocking blocks. The results were subjected to structural simulation for load-carrying capacity and deflection analysis. Finally, in [17], they introduced a computational method to generate planar surface tessellations aimed at TI assembly generation. Their solution is a parametric approach that generates surface tessellations by indicating the number of edges for the tiles and the angle values between the edges. Their system generates a TI configuration (TIC) using a resultant surface tessellation.

Weizmann et al. [18] studied the generation method of TI assemblies, focusing on different block geometries and their influence on structural performance. The study begins with generating two fundamental block shapes—tetrahedral and cubic—derived from regular square and hexagonal tessellations. These tessellations serve as the base for the interlocking blocks, ensuring that each block can interlock with its neighbors through their geometric configuration. A range of assemblies is created by varying the size and inclination angles of the blocks' faces using parametric modeling tools. This involves defining parameters for edge lengths, face angles, and the overall dimensions of the blocks. Based on these parameters, the blocks are arranged into assemblies, forming a grid mimicking potential real-world structures. The generated assemblies are subjected to numerical simulations using the 3DEC software, Version 4.1 which can analyze discontinuous elements and discrete blocks. This simulation includes defining material properties and boundary conditions and applying loads to the assemblies. The software computes the structural response, including displacement and stress distribution. The blocks are often truncated or refined to ensure proper interlocking. This refinement involves modifying the initial shapes to enhance the contact area between blocks, thereby improving the interlocking and load distribution characteristics. The refinement process ensures that the blocks fit together seamlessly without gaps or overlaps. The study conducts a comparative analysis of various assemblies with different geometric configurations. This includes assessing the

impact of block size, face inclination angles, and the overall arrangement on the structural performance. Assemblies are evaluated based on their load-bearing capacity, displacement under load, and failure mode.

Goertzen et al. [19] described a geometric generation method for creating topologically interlocking assemblies using planar crystallographic symmetries. The generation method exploits wallpaper (planar crystallographic) symmetries to create interlocking blocks that can be assembled between two planes. The process begins with selecting a fundamental domain of a planar crystallographic group. This domain represents a regular periodic tessellation of the plane and serves as the base unit for generating interlocking blocks. The fundamental domain is deformed in planes parallel to the original plane. This deformation is carried out using the Escher trick, which involves keeping the boundary points fixed via non-trivial group elements and deforming the boundary connections between these points to retain the fundamental domain property. This results in a new tessellation of the plane with deformed fundamental domains. The deformed fundamental domain is extended into the third dimension by defining a map that continuously deforms one fundamental domain into another across parallel planes. The set of points generated via this deformation forms a polyhedral interlocking block, with each block intersecting planes parallel to the initial plane at different stages of the deformation. The blocks are arranged in a double-periodic fashion based on the symmetries of the chosen planar crystallographic group. This ensures that each block is kinematically constrained by its neighbors, creating a stable interlocking assembly. The assembly process may involve iterating the deformation procedure to generate complex interlocking configurations. A discrete criterion is applied to ensure the stability of the interlocking assembly. This involves verifying that no finite subset of blocks can be moved without causing intersections with other blocks, thereby maintaining the integrity of the interlocking structure.

*2.2. Concave TI Blocks*

Concave block shapes exist that fulfill the TI principle. An example of such shapes is the Osteomorphic Block introduced by Khor et al. [20] and Estrin et al. [21]. More polygonal block styles were reported by Tessmann [22,23], as well as Tessmann and Becker [24]. Such shapes resulted from student projects that built TICs as geometrical differentiated, reversible, force-locked systems. The solutions presented by the students considered a variety of shapes derived from tetrahedra. Although not convex, the planar faces preserve the fundamental notion of the TI principle. Another proposal considers the design of a valid border for a finite assembly. The pieces at the boundary have a special shape that enables the assembly of a frame. Such a solution involves windmill shapes to constrain the boundary, allowing the configuration to end on the edge without an additional peripheral structure.

Akleman et al. [25] proposed a method for topologically interlocking space-filling shapes utilizing Voronoi decomposition based on fabric symmetries. The conceptual framework involves partitioning space using high-dimensional Voronoi sites, such as curves and surfaces, configured according to weave symmetries. This approach guarantees that the generated shapes are space-filling due to the Voronoi-based method and maintains the geometric characteristics essential for topological interlocking in any desired assembly. The method constructs the final Voronoi region by sampling points from the Voronoi sites and computing the union of constitutive Voronoi cells for each sample point. This process allows for the generation of various topologically interlocking shapes called Generalized Abeille Tiles (GATs). The algorithm's simplicity and ability to directly use standard Voronoi cell computation make it a robust method for creating complex interlocking configurations.

Ebert et al. [26] introduced a geometric generation method for creating topologically interlocking configurations using helical structures based on Voronoi tessellation named VoroNoodles. Generating a VoroNoddle involves creating topologically interlocking blocks by tessellating a plane and proliferating the prototype tiles in the third dimension using helical movement. This process combines wallpaper symmetries and Voronoi tessellation to create space-filling, interlocking structures. The method begins by defining helical

trajectories for Voronoi sites. These helices are placed inside an architectured slab, where the Voronoi tessellation is computed at each layer, resulting in a series of tiles created by extruding each layer along the helical paths. Voronoi decomposition is used to partition the space within each layer, ensuring that the resulting blocks are space-filling and that they exhibit strong interlocking properties. The helical trajectories define the Voronoi sites, and each layer's tessellation results in congruent blocks with corrugated boundaries. The method introduces two varieties of helical building blocks—constant cross-sectional noodles and variable cross-sectional noodles. Constant cross-sectional noodles have the same cross-section in each layer, while variable cross-sectional noodles have different cross-sections across layers. This corrugation enhances the interlocking behavior and ensures the structural integrity of the assemblies.

### 2.3. Non-Planar and Curvilinear TI Assemblies

The design of TI assemblies based on non-planar surfaces has been of interest to researchers on the topic.

Vella and Kotnik considered how curvature in the geometric domain affects the resultant pieces of a non-planar Abeille-based TIC [27]. They observed the interdependency between curvature in the geometric domain and the resultant assembly. They found that both curvature and tilting angles are inversely proportional. Additionally, the distribution of curvature and piece vertices is directly proportional (resulting in smaller pieces that approximate the steep curvature).

Weizmann et al. [16] adapted the generation method formulated by Kanel-Belov et al. [3] to work on curvilinear tessellations. Their implementation includes an additional step that adapts the pieces to the curvature of the surface. In such a case, the assembly of the blocks must follow a specific order. Tessmann and Rossi described approaches to designing interlocking pieces based on parametric design logic and discrete combinatorial processes [28,29]. The former returns the geometry of each block according to its location in the tessellation. The latter focuses on exploiting the combinatorial capabilities of repeating block shapes in an assembly. They applied such approaches to generate TICs bounded by two NURBS surfaces. The boundary information helps to generate distorted, trimmed tetrahedra that align with the curvature of the surfaces.

Xu et al. [30] presented a geometric generation method for creating TICs specifically for cylindrical structures. The proposed non-planar interlocking element was designed with a symmetrical geometry featuring six concave–convex side surfaces, facilitating interlocking with adjacent elements. Creating an element starts with a regular hexagon and its circumcircle. Then, three non-adjacent arcs are reflected. These arcs are connected to form the top surface of the planar interlocking element, and the bottom surface is obtained by rotating the same shape by 60 degrees. The respective side surfaces are created by lofting the edges of the top and bottom surfaces. Finally, the planar interlocking element is morphed to fit the target cylindrical surface using a "*flow along surface*" technique. This transformation results in a non-planar interlocking element suitable for assembling tubular structures. The assembled structure, consisting of these elements, displays an identical pattern on the inner and outer surfaces. Each element is mechanically interlocked with its six neighbors, restricting movement in all directions purely through geometric constraints.

Kurucu and İlerisoy [31] proposed creating TI flat vaults using truncated octahedra with enhanced joint details to improve structural performance. The process begins with creating a hexagonal tessellation. Truncated octahedra are generated from this tessellation using the Starfish plug-in for Grasshopper. This plug-in facilitates the creation of parametric patterns, which are essential for generating the base geometric forms. X-joints are added to the truncated octahedra to enhance the contact surface area and improve structural performance. These X-joints are incorporated through volumetric modifications using Rhino software. Each X-joint increases the contact surface area by approximately ten $cm^2$, significantly improving the structural stability of the interlocking blocks. Two flat vault configurations are designed—one using truncated octahedra and the other using truncated

octahedra with X-joints. The design process involves ensuring that the interlocking edges of the blocks fit together seamlessly, providing a robust and stable assembly. The structural performance of the designed flat vaults is analyzed using finite element analysis (FEA) in SimScale. The FEA assesses the displacement and von Mises stress distributions within the vaults under applied loads. This analysis helps quantify the improvements in structural performance due to the addition of X-joints.

*2.4. Free-Form TI Assemblies*

Recent TI generation approaches aim to generalize the process for assemblies based on free-form 3D surface tessellations.

Bejarano and Hoffmann [7] proposed the Height–Bisection method based on height parameters. The traditional generation method designs the interlocking pieces by assigning a tilting plane to each tile edge, defined by a tilting angle from the tessellation plane. This plane intersects with adjacent planes to form the vertices and edges of the polyhedra. The traditional method, however, requires multiple iterations to adjust the tilting angles to ensure no overlapping occurs, especially in non-planar or irregular tessellations. In contrast, the proposed Height–Bisection method simplifies the process by using a height parameter and a central point within each face, eliminating the need for iterative angle adjustments. The Height–Bisection method calculates rotation vectors for each edge based on these parameters, generating interlocking pieces in a single iteration. This method aligns the equatorial sections of the pieces with the planes of the tessellation faces, significantly reducing overlaps and ensuring proper alignment in the resulting configuration.

Wang et al. [32] considered the case of global interlocking to generate and adapt the assembly blocks. Their approach adapts the shape of the blocks as the procedure tests the static equilibrium analysis of the assembly at multiple orientations until no block can be displaced from its location.

Loing et al. [33] presented a generation method for creating a TIC modeled after a free-form and curved surface. The generation method utilizes a parametric design approach based on planar quadrilateral meshes representing the desired surface. Then, a moving cross-section procedure is employed to ensure the elements are interlocked. This involves defining a set of tilting angles for each edge of the quadrilateral facets, which helps create interlocked systems constituted by blocks such as regular tetrahedrons. The tilting angles are varied to match the curvature of the surface. A mathematical criterion based on a set of inequalities is applied to guarantee the interlocking property of the blocks. These inequalities ensure that each block is translationally locked, meaning it cannot be removed by translation if its neighbors are fixed. The criterion involves checking the normal vectors of the planes defining each block to ensure that they satisfy the interlocking conditions. Once the initial blocks are generated, they may need to be truncated or refined to fit perfectly into the mesh and accommodate overlaps. This step ensures that the blocks have proper face-to-face contact, enhancing the structural integrity of the assembly.

Chen et al. [34] proposed a method that models free-form shell structures by parameterizing the geometry of shell elements and clustering them into discrete equivalence classes to optimize reusability and buildability. The process begins with a 3D freeform surface remeshed to obtain a base polygonal mesh. This base mesh is created by mapping a 2D tessellation with convex polygons onto the surface using the as-rigid-as-possible algorithm. The tessellation is chosen to be monohedral, dihedral, or trihedral to limit the number of distinct tiles. Each polygonal face in the base mesh is augmented with vectors orthogonal to its edges. These augmented vectors are used to define 3D partitioning planes, which construct the convex geometry of the corresponding shell elements. The side faces of each shell element are generated based on these partitioning planes. The shell elements are clustered based on their geometric similarity, defined by edge lengths, diagonal lengths, and augmented angles. A hierarchical clustering approach is employed to group elements into discrete equivalence classes. The clustering process aims to balance the reusability of the template elements with the seamlessness and buildability of the final structure. Three error

metrics are defined to ensure that the final structure is both seamless and buildable: the contact error, gap error, and overlap error. These metrics measure the deviations between the shell elements' side faces and their planar contacts, ensuring minimal gaps and overlaps. For each cluster of shell elements, a template element is generated by averaging the vertices of the clustered elements. The templates replace the original elements, creating a shell structure composed of limited, unique shapes. The base mesh and the augmented angles are optimized hierarchically to minimize the number of templates while preserving the structure's integrity. The optimization process adjusts the vertices and augmented angles to reduce the clustering errors and improve the geometric fit of the shell elements.

Laccone et al. [35] presented a computational design methodology for segmented concrete shells constructed from post-tensioned precast flat tiles. The geometric generation method involves decomposing the target shell shape into flat quadrilateral tiles, each forming right prisms with sides and faces at 90-degree angles. The segmentation process ensures that the tiles touch at the midpoint of their edges, creating a shell that remains compressed when assembled. The design pipeline begins with an input shape and progresses through steps including tile genesis, cable path optimization, and structural modeling. A tailored algorithm facilitates this process, transforming the input shape into a detailed design of prismatic tiles with embedded cable ducts and steel segments at the contact interfaces. The tiles are post-tensioned post-assembly to minimize tensile forces under service load, enhancing the shell's structural integrity. The final assembly includes an in situ cast to fill gaps and activate the shell behavior, ultimately improving bending strength and maintaining an efficient force flow while achieving the desired aesthetic.

## 3. Polygon Evolution

This section describes the generation of a polyhedron from a polygon using multiple evolution steps. An evolution step is a sweeping procedure that evolves (i.e., reshapes and translates) a polygon into a $n$-polytope (i.e., a generalized polyhedron in the $n$-th dimension), where $n = 1, 2, 3$. The possible resultant evolved shapes are points, line segments, or polygons. This method is a sweep-plane algorithm along a direction vector. The vertices from a sequence of evolved $n$-polytopes are the vertices of the resultant polyhedron. The continuous motion of the polygon, as it evolves to the respective $n$-polytope, describes the edges of the polyhedron.

### 3.1. Evolution Step

An evolution step is a process that reshapes and translates a polygon into a generalized polytope. We show, in Figure 3, the stages of the evolution process of a polygon, $f$, into a polygon, $f'$. The evolution step requires a couple of parameters: a set of angles, $\Theta$, and a distance value, $\lambda$. $\Theta$ indicates the rotation of the normal vector of $f$ when placed at each one of its edges. $\lambda$ indicates the far plane at which $f'$ can be translated the most. The evolution process calculates incident planes to each edge of $f$ based on the translated normal vectors and the respective edge midpoints. The intersection of such planes defines a set of lines, $L$, whose intersection with one another or with the far plane defines the vertices of the evolved polygon, $f'$. We state this process formally as $f = \text{Evolve}(f, \Theta, \lambda)$. The union of the vertices from $f$ and $f'$ describes the vertices of the respective evolving solid. For this particular example, the resulting polyhedron is the top half of a regular octahedron.

The following paragraphs describe the polygon evolution process in formal terms. Let $f = \{V, E\}$ be a polygon of $n \in \mathbb{N}^+, n \geq 3$ sides, where $V = \{v_0, v_1, \ldots, v_{n-1}\}$ is a set of vertices, and $E = \{(0, 1), (1, 2), \ldots, (n-2, n-1), (n-1, 0)\}$ is the set of tuples with the vertex indices that define the edges of $f$. The polygon has centroid point $C$ and normalized normal vector $\hat{N}$. We assume that $V$ and $E$ walk around $f$ in counterclockwise order to $C$ and $\hat{N}$. Each edge, $e = (i, j) \in E$, has a normalized direction vector, $\hat{K}_e = \frac{v_j - v_i}{||v_j - v_i||}$, and midpoint, $M_e = \frac{v_i + v_j}{2}$.

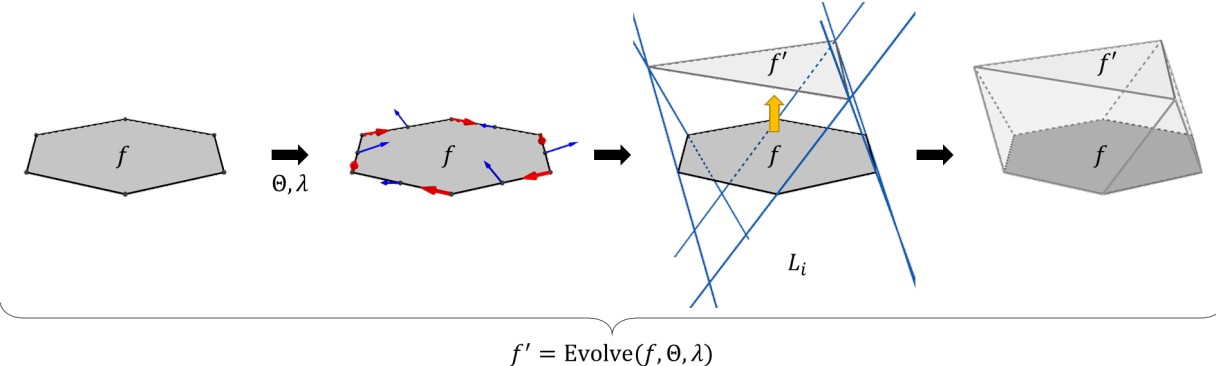

$$f' = \text{Evolve}(f, \Theta, \lambda)$$

**Figure 3.** Example of the stages for evolving a polygon into a *n*-polytope. From **left** to **right**: seed polygon, evolution parameters (blue arrows represent the edge tilted normal vectors, red arrows represent the edge direction vectors), definition of the evolved polytope from the intersection of evolving lines (yellow arrow indicates the evolution direction), and the resultant polyhedron by joining the seed polygon and the evolved polytope.

An evolution step requires two types of parameters: tilting angles associated with the edges of the polygon and a scalar value determining the maximum length allowed for the step. Let $\Theta = \{\theta_e, \forall e \in E \mid \theta_e \in [-\frac{\pi}{2}, \frac{\pi}{2}]\}$ be a set of tilting angles, and let $\lambda \in \mathbb{R}^+$ be a scalar value. A tilting angle, $\theta_e$, rotates $\hat{N}$ using $\hat{K}_e$ as the rotation axis vector; the rotated vector $\hat{N}_e$ is the normal vector of a plane that contains $e$. Then, $\hat{N}_e = \text{rotate}(\hat{N}, \hat{K}_e, \theta_e)$, where $\text{rotate}(\hat{N}, \hat{K}_e, \theta_e)$ is the axis–angle rotation (also known as Rodrigues's rotation formula). The plane $P_e = \text{plane}(M_e, \hat{N}_e)$ passes through $e$. The intersection between the planes from the two edges incident to a vertex, $v_i \in V, \forall i = 0, 1, \ldots, n-1$, defines a line, $L_i$; therefore, $v_i \in L_i$. Finally, a point, $D = C + \lambda \hat{N}$, and planes $P_C = \text{plane}(C, \hat{N})$ and $P_D = \text{plane}(D, \hat{N})$. The top right subfigure in Figure 4 shows an example of the mentioned elements on a square polygon.

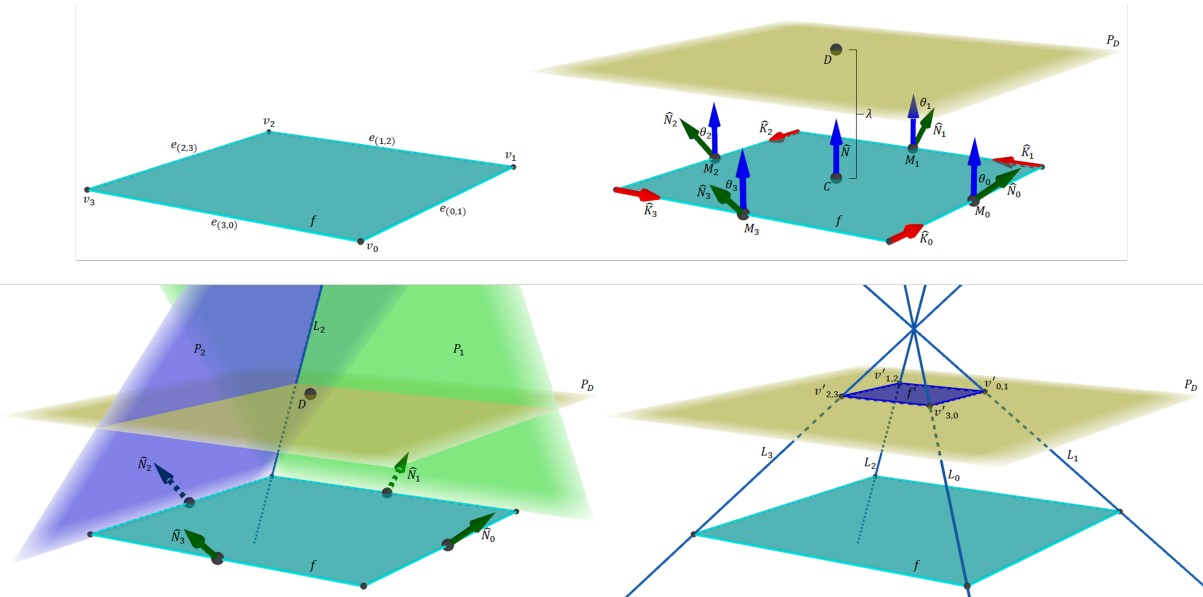

**Figure 4.** Detailed illustration of the elements required to evolve a square polygon, $f$, into a polygon, $f'$. (**Top left**): seed square polygon. (**Top right**): rotated normal vectors and plane $P_D$. (**Bottom left**): incident planes to the edges of $f$. (**Bottom right**): evolved polygon $f'$ defined as the intersection points between lines $L_i$ and the plane $P_D$.

We consider the intersection points $v'_{i,j} = L_i \cap L_j, \forall (i, j) \in E$ that occur within the space section delimited by planes $P_C$ and $P_D$. A point, $v'_{i,j}$, is within evolution range if

$v'_{i,j} \in P_C^+ \cap (P_D^- \cup P_D)$. That is, $v'_{i,j}$ lies at both the positive half-space defined by plane $P_C$ in direction $\hat{N}$ and the negative half-space defined by plane $P_D$ in the opposite direction to $\hat{N}$ or $P_D$ itself. Otherwise, $v'_{i,j}$ is out of the evolution range. In such case, we consider the intersection points $v'_{i,D} = L_i \cap P_D$ and $v'_{j,D} = L_j \cap P_D$, which are within evolution range. The bottom-right subfigure in Figure 4 shows an evolved $n$-polytope $f'$ defined by the intersection points between lines $L_i$ and the plane $P_D$. Let $V'$ be the set of intersection points within the evolution range; then, $V'$ defines the endpoints of the evolved $n$-polytope $f'$. Let $E'$ be the set of edges of $f'$. Each tuple, $e' \in E'$, is defined by the indices of consecutive vertices in $V'$ as they are calculated. The last tuple in $E'$ must connect the last and the first vertices in $V'$. We call $f' = \text{EVOLVE}(f, \Theta, \lambda)$ the described procedure that evolves $f = \{V, E\}$ into $f' = \{V', E'\}$. We refer to $f$ as the seed polygon when used as a parameter in an evolution step.

Algorithm 1 shows the pseudocode for the EVOLVE procedure. The algorithm requires three traversals through the $n$ edges of the seed polygon: The first loop defines the planes incident to each edge. The second loop calculates the line segments, incident to each vertex, representing the intersection of the respective incident edges. The third loop calculates the vertices of the evolved polygon, $f'$, by checking the intersection between the respective line segments per edge and its location regarding the polygon plane and the evolution plane. Each loop structure has $n$ iterations, resulting in EVOLVE $\in O(n)$.

The cardinality of $V'$ determines the type of $n$-polytope of $f'$. Polytope $f'$ is a single point if $|V'| = 1$, a line segment if $|V'| = 2$, or a polygon with $n' = |V'|$ sides if $|V'| \geq 3$. When $|V'| > 3$, we need to check whether all points in $V'$ are coplanar. If at least one point $v'_i \in V'$ is not coplanar, then $f'$ is a degenerated evolution of $f$ (i.e., the sequence of edges $E' = \{(0,1), (1,2), \ldots, (n'-2, n'-1), (n'-1, 0)\}$ does not describe a planar polygon). In such a case, adjusting either $f$, $\Theta$, or $\lambda$ such that the resultant $n$-polytope $f' = \{V', E'\}$ describes either a point, a line segment, or a planar polygon solves the issue. Figure 5. shows examples of evolved $n$-polytopes from regular seed polygons.

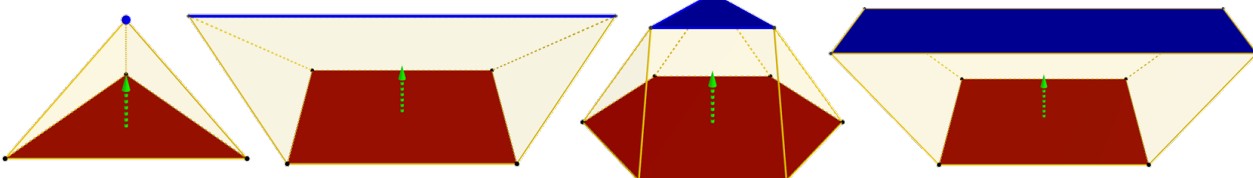

**Figure 5.** Evolved $n$-polytopes from polygons. Seed polygons are in red, and evolved $n$-polytopes are in blue. From **left** to **right**: point, line segment, polygon with fewer sides than the seed polygon, and polygon with the same number of sides as the seed polygon.

An evolution step generates the geometry of a prismatoid whose vertices are $\{V \cup V'\}$. Such a set of vertices, along with the set of tilting angles $\Theta$, describes the type of the generated prismatoid. Table 1 shows the requirements for vertices and angles to describe a specific polyhedron from the prismatoids family.

**Table 1.** Prismatoid families based on the original vertices, $V$, evolved vertices, $V'$, and tilting angle parameters, $\Theta$, using a single evolution step.

| Family | Vertices | Angles |
|---|---|---|
| Pyramids | $|V| \geq 3, |V'| = 1$ | $\theta > 0, \forall \theta \in \Theta$ |
| Wedges | $|V| \geq 3, |V'| = 2$ | $\theta > 0, \forall \theta \in \Theta$ |
| Parallelepipeds | $|V| = |V'| = 4$ | $\theta_0 = \theta_2 = 0, \theta_1 = \theta_3$ |
| Prisms | $|V| = |V'| \geq 3$ | $\theta = 0, \forall \theta \in \Theta$ |
| Cupolae | $|V| > |V'| \geq 3$ | $\theta > 0, \forall \theta \in \Theta$ |
| Frusta | $|V| = |V'| \geq 3$ | $\theta > 0, \forall \theta \in \Theta$ |

---

**Algorithm 1** Evolve algorithm.

---

 1: **function** EVOLVE($f$ : polygon with $n$ sides, $\Theta$ : list of angles, $\lambda : \mathbb{Z}$)
 2:     let $D\{V, F, H\}$ be the DCEL representing $f$
 3:     $h \leftarrow F[0].\text{halfedge}$                                      ▷ $D$ has only one face
 4:     **do**                                          ▷ Set the planes incident to the half edges
 5:         $h.\text{plane} \leftarrow \text{PLANE}(h.\text{midpoint}(), \text{ROTATE}(h.\text{normal}(), h.\text{direction}(), \Theta_i))$
 6:         $h \leftarrow h.\text{next}$
 7:     **while** $h \neq F[0].\text{halfedge}$
 8:     **do**                           ▷ Calculate the intersections between half-edge planes
 9:         $v \leftarrow h.\text{start}$
10:         $v.L \leftarrow \text{INTERSECT}(h.\text{previous.plane}, h.\text{plane})$
11:         $h \leftarrow h.\text{next}$
12:     **while** $h \neq F[0].\text{halfedge}$
13:     $\hat{N} \leftarrow ||F[0].\text{normal}()||$
14:     $C \leftarrow F[0].\text{centroid}()$
15:     $D \leftarrow C + \lambda \hat{N}$
16:     $P_C \leftarrow \text{PLANE}(C, \hat{N})$
17:     $P_D \leftarrow \text{PLANE}(D, \hat{N})$
18:     let $V'$ be an empty collection of unique points
19:     **do**                                    ▷ Calculate the intersections between line segments
20:         $v_0 \leftarrow h.\text{start}$
21:         $v_1 \leftarrow h.\text{twin.start}$
22:         $v' \leftarrow \text{INTERSECT}(v_0.L, v_1.L)$
23:         **if** $v'$ exists, **then**
24:             **if** $\hat{N} \cdot (v' - D) > 0$, **then**                              ▷ $v'$ above $P_D$
25:                 $V'.\text{append}(\text{INTERSECT}(P_D, v_0.L))$
26:                 $V'.\text{append}(\text{INTERSECT}(P_D, v_1.L))$
27:             **else if** $\hat{N} \cdot (v' - D) = 0$, **then**                              ▷ $v'$ at $P_D$
28:                 $V'.\text{append}(v')$
29:             **else if** $\hat{N} \cdot (v' - D) < 0$ **and** $\hat{N} \cdot (v' - C) > 0$ , **then** ▷ $v'$ between $P_D$ and $P_C$
30:                 $V'.\text{append}(v')$
31:             **else**                                          ▷ $v'$ below $P_C$
32:                 $V'.\text{append}(\text{INTERSECT}(P_D, v_0.L))$
33:                 $V'.\text{append}(\text{INTERSECT}(P_D, v_1.L))$
34:         **else**                                      ▷ Line segments are parallel
35:             $V'.\text{append}(\text{INTERSECT}(P_D, v_0.L))$
36:             $V'.\text{append}(\text{INTERSECT}(P_D, v_1.L))$
37:         **end if**
38:         $h \leftarrow h.\text{next}$
39:     **while** $h \neq F[0].\text{halfedge}$
40:     **return** $V'$
41: **end function**

---

### 3.2. Single-Direction Polygon–Polyhedron Evolution

It is possible to generate a more elaborated polyhedron by applying a sequence of multiple evolution steps. In this case, an evolved polygon serves as the seed polygon for another evolution step with its own set of parameters. Figure 6 shows an example of an evolution step with a seed polygon, $f'$, obtained from a previous evolution step. A newly evolved polytope, $f'' = \text{EVOLVE}(f', \Theta', \lambda')$, corresponds to the second evolution step in a single-direction sequence. In such an example, the union of the vertices from polygons $f$, $f'$, and $f''$ represents a new polyhedron.

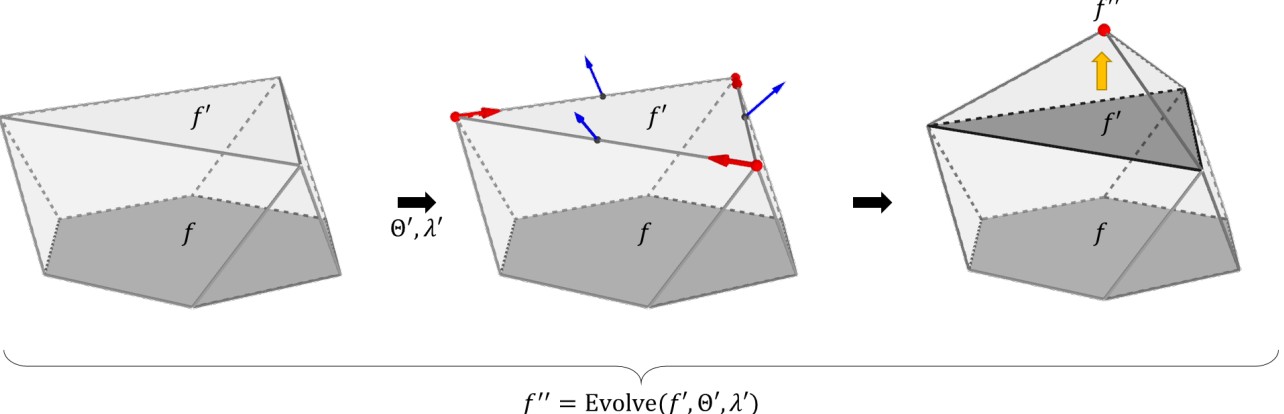

$$f'' = \text{Evolve}(f', \Theta', \lambda')$$

**Figure 6.** Example of the stages of a single-direction polygon–polyhedron evolution. From **left** to **right**: initial evolution step from $f$ to $f''$, evolution parameters (blue arrows represent the edge tilted normal vectors, red arrows represent the edge direction vectors), evolved point $f''$, and the resultant polyhedron as the union of the vertices from $f$, $f'$, and $f''$ (yellow arrow indicates the evolution direction).

The following paragraphs describe the single-direction polygon–polyhedron evolution process in formal terms. We consider using evolved $n$-polytopes as the seed polygons for subsequent evolution steps. An evolved $n$-polytope $f' = \text{EVOLVE}(f, \Theta, \lambda)$ where $f' = \{V', E'\}$ with $|V'| \geq 3$ becomes the seed polygon for another evolution step. Such a new step requires its own set of evolution parameters, $\{\Theta', \lambda'\}$ with $|\Theta'| = |V'|$ (i.e., the number of tilting angles for the new evolution step is equal to the number of sides of $f'$). A new evolved $n$-polytope $f'' = \text{EVOLVE}(f', \Theta', \lambda')$ is, then, an additional step in the evolution sequence from the original seed polygon, $f = \{V, E\}$, into a $n$-polytope, $f'' = \{V'', E''\}$. Considering the set of vertices $\{V, V', V''\}$ from polygons $f, f'$ and $n$-polytope $f''$, this set contains the vertices of a polyhedron described by the two-step evolution of $f$ into $f''$. Figure 7 shows an example of the elements for the evolution of a polygon into a single point through two evolution steps.

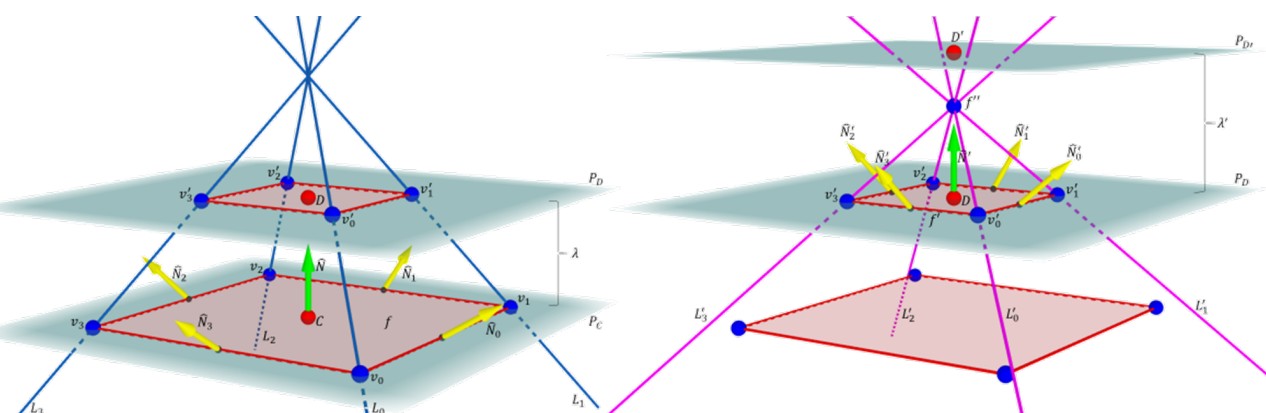

**Figure 7.** Detailed illustration of the elements required for a single-direction polygon0-polyhedron evolution. (**Left**): elements for the first evolution step from a square to a single point. (**Right**): elements for a second evolution step from a square to a single point.

A sequence of, at most, $m \geq 2$ evolution steps generates a polyhedron, $B$, from a polygon, $f$. Let $f^0 = f$ be the seed polygon, the sequence of evolved $n$-polytopes $f^{i+1} = \text{EVOLVE}(f^i, \Theta^i, \lambda^i)$, where $f^i = \{V^i, E^i\} \, \forall i = 0, 1, \ldots, n - 1$ contains the geometric information of $B$. The evolution process stops after $m$ evolution steps, or a $n$-polytope $f^i$ is either a point or a line segment. The set $V_B = \bigcup_{i=0}^{m-1} V^i$ is the set of vertices of $B$. The set $F_B$

is the set of vertex indices that defines the faces of $B$. The correspondences between the edges of $f^i$ and the respective evolved elements from $f^{i+1}$ define $F_B$.

### 3.3. Double-Direction Polygon–Polyhedron Evolution

We expand the family of evolved polyhedra by considering negative evolution steps. A negative evolution step is the opposite of the normal vector from the respective seed polygon. We must flip the orientation of the seed polygon for the first negative evolution. In that way, we can consider negative evolution steps as regular evolution steps, as discussed so far. Figure 8 shows an example of the stages during a double-direction polygon–polyhedron evolution process. The polygon $f$ is the seed polygon for two evolution sequences: a positive (along the normal vector of $f$) step and a negative step (opposite to the normal vector of $f$). We refer to $f$ as $f_0^+$ for the positive step, and $f_0^-$ for the negative step. A new evolved polytope $f_1^- = \text{EVOLVE}(f_0^-, \Theta_0^-, \lambda_0^-)$ corresponds to the negative evolution step. In such an example, the union of the vertices from polygons $f$, $f_1^+$ and $f_1^-$ represents a new polyhedron.

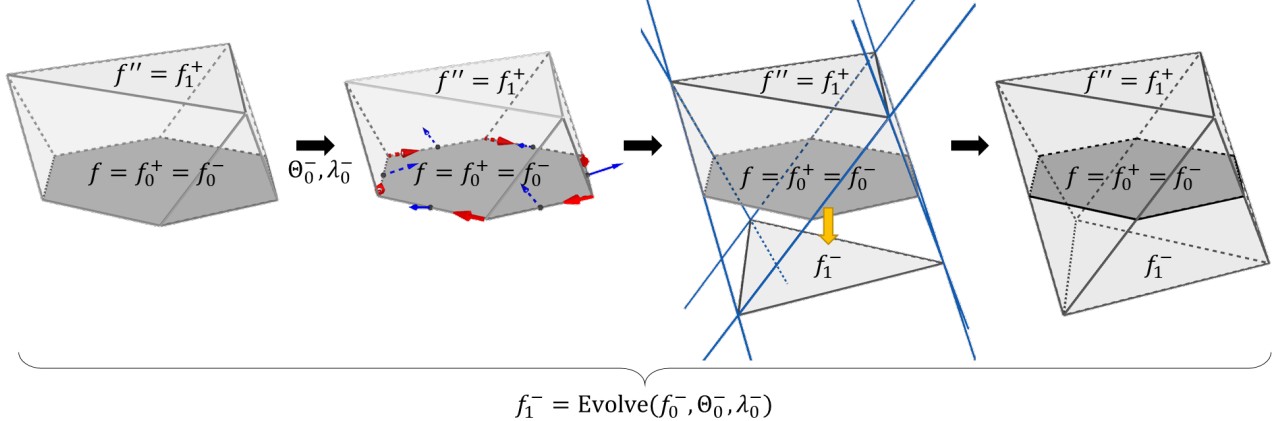

$$f_1^- = \text{Evolve}(f_0^-, \Theta_0^-, \lambda_0^-)$$

**Figure 8.** Example of the stages of a double-direction polygon–polyhedron evolution. From **left** to **right**: Initial evolution step from $f$ to $f'' = f_1^+$ (positive evolution step), Parameters for the negative evolution step, Definition of the evolved polytope from the intersection of evolving lines resulting from the negative evolution step, Resultant polyhedron by joining the vertices from $f$, $f_1^+$, and $f_1^-$.

The following paragraphs describe the double-direction polygon–polyhedron evolution process in formal terms. Let $f$ be a seed polygon with a normalized normal vector, $\hat{N}$. An additional sequence of evolution steps, along with $-\hat{N}$, allows for the generation of polyhedra that a single-direction polygon–polyhedron evolution cannot describe. A positive evolution step occurs above $f$ along $\hat{N}$, and a negative evolution step occurs below $f$ along $-\hat{N}$.

Let $f_0^+ = f$ be the seed polygon for the first positive evolution step; the sequence of $m^+ \in \mathbb{N}^+$ positive evolution steps $f_{i+1}^+ = \text{EVOLVE}(f_i^+, \Theta_i^+, \lambda_i^+)$, $i = 0, 1, \ldots, m^+ - 1$ describes the upper half of the polyhedron (i.e., the section that lies above $f$ in direction $\hat{N}$). Similarly, let $f_0^- = \text{flip}(f)$ be the seed polygon for the first negative evolution step, where $\text{flip}(p)$ changes the vertex indices order of the edges from a polygon, $p$ (i.e., swaps the orientation of the front face of the polygon). Flipping the seed polygon for the first negative evolution step allows for the evolution of $f$ along $-\hat{N}$. This adjustment lets us use the evolution step method without any modification for negative evolution steps. The sequence of $m^- \in \mathbb{N}^+$ negative evolution steps $f_{j+1}^- = \text{EVOLVE}(f_j^-, \Theta_j^-, \lambda_j^-)$, $j = 0, 1, \ldots, m^- - 1$ describes the lower half of the polyhedron (i.e., the section that lies below $f$ in direction $-\hat{N}$).

The geometry of an evolved polyhedron, $B$, comes from the vertices of the seed polygon and the evolved $n$-polytopes, along with the positive and negative directions. The vertices from the seed polygon $f = \{V, E\}$, the positive evolved $n$-polytopes $f_i^+ = \{V_i^+, E_i^+\}, \forall i = 0, 1, \ldots, m^+ - 1$, and the negative evolved $n$-polytopes $f_j^- = \{V_j^-, E_j^-\}$,

$\forall j = 0, 1, \ldots, m^- - 1$ describe the vertices of the polyhedron $B$ generated from both positive and negative evolution sequences. The set $V_B = V \cup \bigcup_{i=0}^{m^+ - 1} V_i^+ \cup \bigcup_{j=0}^{m^- - 1} V_j^-$ contains the vertices of $B$. The set $F_B$ is the set of vertex indices that defines the faces of $B$. Similar to the single-direction evolution, the correspondences between the edges of the seed polygons and their respective evolved $n$-polytopes define $F_B$. Figure 9b shows an example of a polyhedron obtained from evolving a seed polygon, $f$, along $\hat{N}$ and $-\hat{N}$.

By omitting the vertices of the seed polygon, $f$, we can generate a different polyhedron, $B$, using only the information from both evolution sequences. The set $V_B = \bigcup_{i=0}^{m^+ - 1} V_i^+ \cup \bigcup_{j=0}^{m^- - 1} V_j^-$ is, then, the vertices from both positive and negative evolved $n$-polytopes exclusively. Figure 9c shows an example of a polyhedron obtained from evolving a seed polygon, $f$, along $\hat{N}$ and $-\hat{N}$ without including the vertices of $f$.

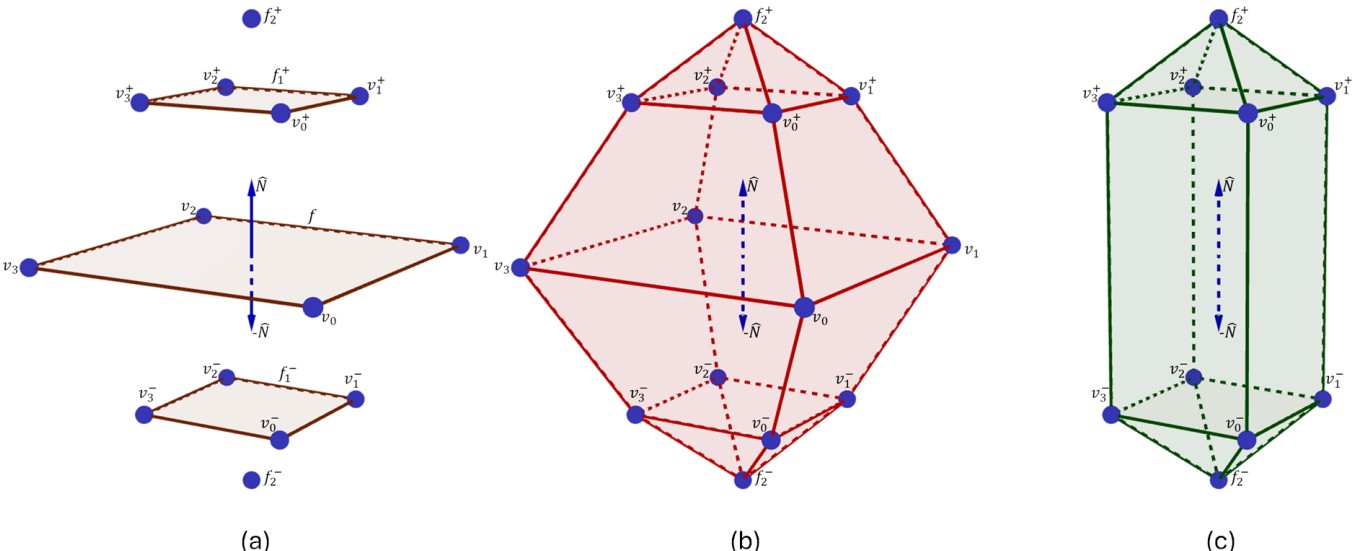

(a)                    (b)                    (c)

**Figure 9.** Resultant polyhedra from double-direction polygon–polyhedron evolution from a polygon, $f$. (**a**): seed polygon and $n$-polytopes from evolution steps, (**b**): polyhedron described by vertices from $f, f_1^+, f_2^+, f_1^-, f_2^-$, (**c**): polyhedron described by vertices from $f_1^+, f_2^+, f_1^-, f_2^-$.

### 3.4. Reciprocal Evolution Steps

So far, we can generate polyhedra with coplanar faces between positive and negative evolution steps. However, that is only possible with adequate evolution parameters. We introduce the idea of reciprocal evolution steps to guarantee coplanar faces between the first positive and negative evolution steps. We say two evolution steps are reciprocal if one is a "*mirror*" version of the other. One evolution step is a $\pi$ rotation of the other. Such evolution steps provide a mechanism to maintain coplanar faces between the first positive and negative evolution steps, provided their respective parameters are reciprocal.

Figure 10 shows an example of a positive evolution step and its reciprocal negative evolution step. In the example, the reciprocal evolution guarantees the linearity of the faces of the resulting polyhedron between the first positive and negative steps. We require such a feature to describe well-known polyhedra such as the Platonic solids.

We consider rotational symmetries between evolved $n$-polytopes on double-direction polygon–polyhedron evolution. Let $f = \{V, E\}$ be a polygon and seed polygons $f_0^+ = f$, $f_0^- = \text{flip}(f)$ for both positive and negative evolution sequences, respectively. A positive evolved $n$-polytope, $f_{i+1}^+ = \text{EVOLVE}(f_i^+, \Theta_i^+, \lambda_i^+), i \in \{0, 1, \ldots, m^+ - 1\}$, is reciprocal to a negative evolved $n$-polytope, $f_{j+1}^- = \text{EVOLVE}(f_j^-, \Theta_j^-, \lambda_j^-), j \in \{0, 1, \ldots, m^- - 1\}$, if $|V_i^+| = |V_j^-|$, $|V_i^+|$ and $|V_j^-|$ are even numbers, and the geometries of $f_{i+1}^+$ and $f_{j+1}^-$ are rotational symmetric to one another of order $O = \frac{|V_i^+|}{2}$. Otherwise, the evolved $n$-polytopes are non-reciprocal to each other. Finally, two sets of parameters, $\{\Theta_i^+, \lambda_i^+\}, \{\Theta_j^-, \lambda_j^-\}$ for

$i \in \{0, 1, \ldots, m^+ - 1\}$ and $j \in \{0, 1, \ldots, m^- - 1\}$, are reciprocal if the respective evolved $n$-polytopes $f_{i+1}^+, f_{j+1}^-$ are reciprocal to each other. Figure 11 shows two octahedra generated using reciprocal and non-reciprocal evolution steps.

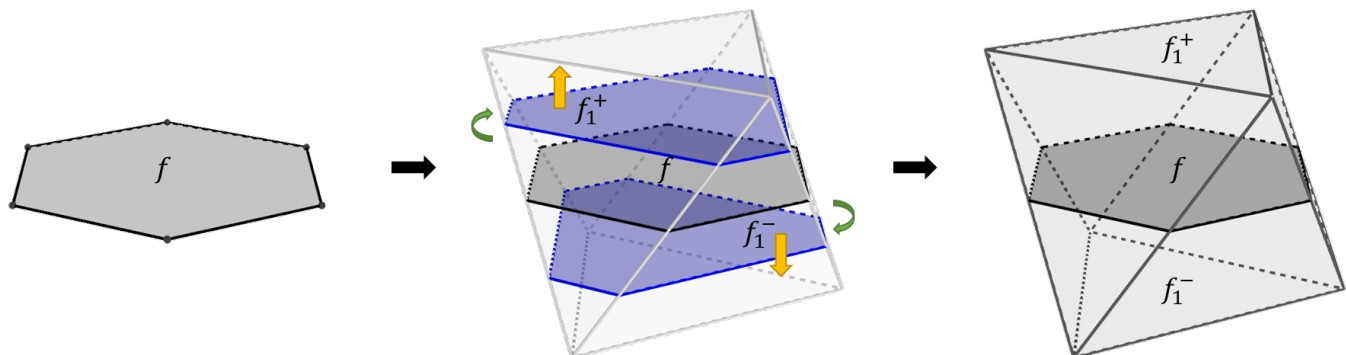

**Figure 10.** Example of the stages of a reciprocal evolution. From **left** to **right**: seed polygon (hexagon) $f$, positive evolution step $f_1^+$ and reciprocal negative evolution step $f_1^-$, and the resultant polyhedron (regular octahedron) by joining the vertices from $f$, $f_1^+$, and $f_1^-$.

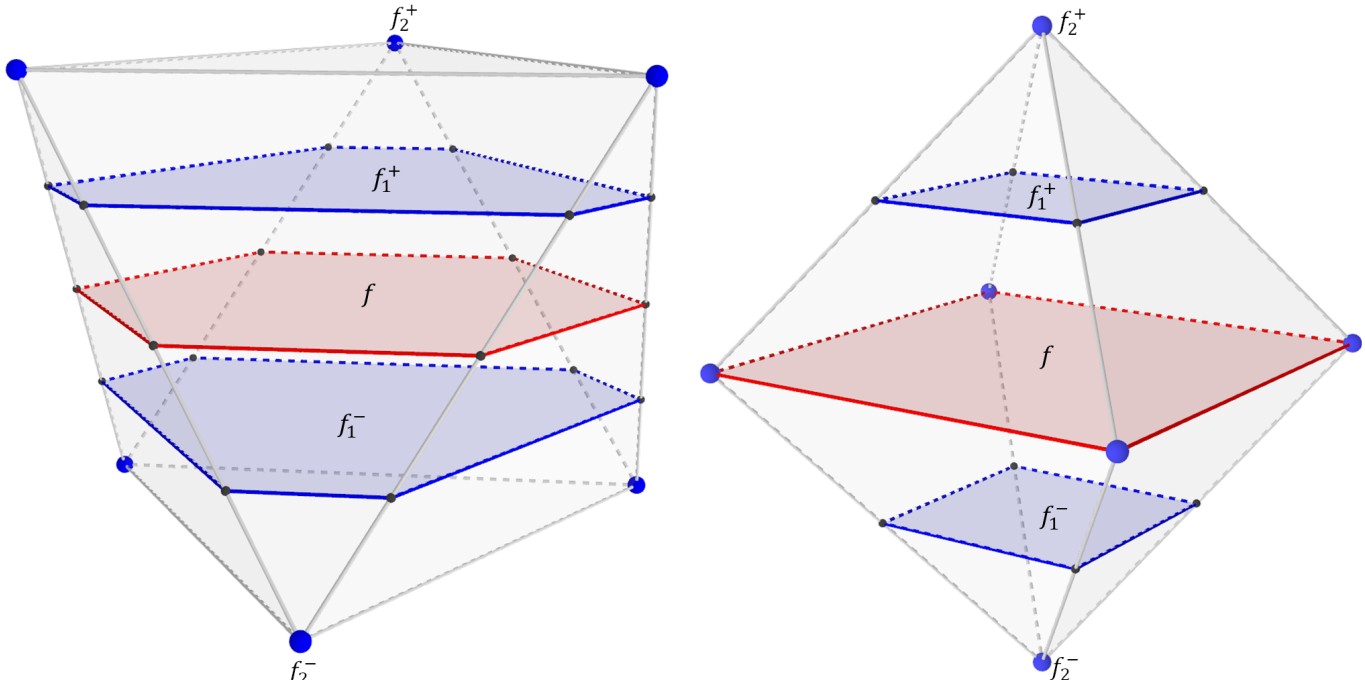

**Figure 11.** Evolved octahedra based on reciprocal and non-reciprocal evolution steps. (**Left**): reciprocal evolution steps $f_1^+$ with $f_1^-$, and $f_2^+$ with $f_2^-$, (**Right**): non-reciprocal evolution steps.

### 3.5. Uniform Evolution

The double-direction evolution through a sequence of correspondent reciprocal evolution steps between positive and negative evolution sequences generates a different set of polyhedra. Let $f = \{V, E\}$ be a polygon, seed polygons $f_0^+ = f$, $f_0^- = \mathrm{flip}(f)$ for both positive and negative evolution sequences, respectively, a positive evolution sequence with $m^+$ evolution steps and a negative evolution sequence with $m^-$ evolution steps. Both evolution sequences are uniform evolutions of $f$ if $m^+ = m^-$ (i.e., the number of evolution steps for both positive and negative evolution sequences is the same) and the evolved $n$-polytopes $f_{i+1}^+ = \mathrm{EVOLVE}(f_i^+, \Theta_i^+, \lambda_i^+)$ and $f_{i+1}^- = \mathrm{EVOLVE}(f_i^-, \Theta_i^-, \lambda_i^-)$ are reciprocal to each other $\forall i = 0, 1, \ldots, m^+ - 1$. Figure 12 shows two polyhedra generated with uniform and non-uniform evolution. The uniform evolution generates a truncated tetrahedron.

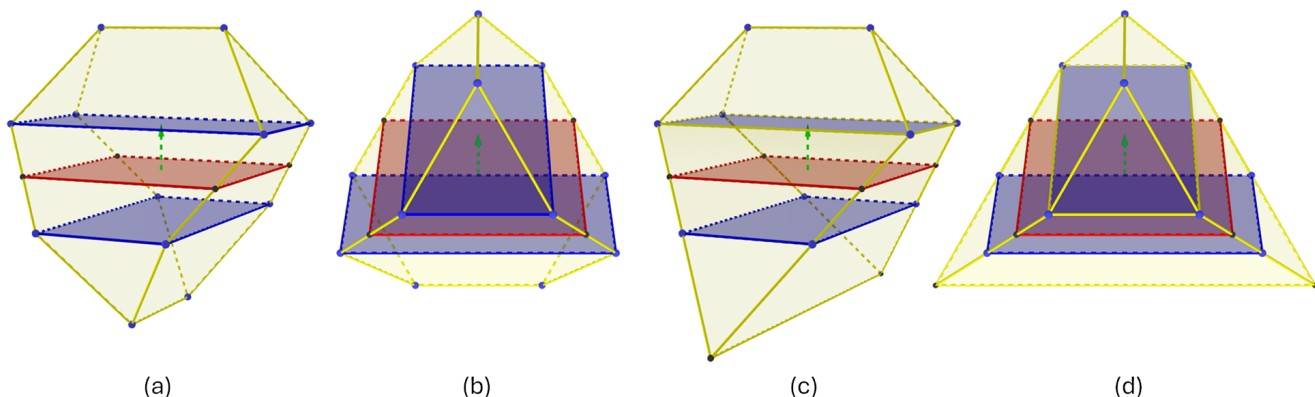

**Figure 12.** Uniform and non-uniform evolution. Seed polygons are in red. Evolution polygons are in blue. (**a**) uniform evolution (overview), (**b**) uniform evolution (lateral), (**c**) non-uniform evolution (overview), and (**d**) non-uniform evolution (lateral).

## 4. General Mid-Section Evolution

We propose an evolutionary approach that defines the blocks of a TI assembly based on a given surface tessellation. Each tile in the tessellation evolves into a polyhedron via a double-direction evolution. Although each tile evolves independently from the others, the parameters of at least its first positive and negative evolution steps must match with the respective first step parameters from the neighboring tiles. Such a requirement guarantees the evolved polyhedra have common interfaces that allow them to comply with the TI principle (to be discussed in Section 4.2). The evolved polyhedra are, then, the blocks of the interlocking assembly.

### 4.1. Generation Method

Let $M = \{V, F\}$ be a surface tessellation with $V$ being the set of vertices and $F$ being the set of tiles. Each tile, $f \in F$, must have an even number of sides (this guarantees the edges will have alternating directional values, as indicated in Section 3.4). A tile, $f \in F$, evolves into a polyhedron, $B$, which becomes the respective block in the TI assembly. The geometry of $B$ comes from the double-direction polygon–polyhedron evolution of $f$ with $f_0^+ = f$ and $f_0^- = \text{flip}(f)$ being the respective seed polygons for the positive and negative evolution sequences, respectively. Each evolution sequence has a specific number of steps. The positive sequence has $m^+ \in \mathbb{N}^+$ steps, while the negative sequence has $m^- \in \mathbb{N}^+$ steps.

An evolution step takes a set of evolution parameters, $\Psi_k^d = \{\Theta_k^d, \lambda_k^d\}$, $\forall k = 0, 1, \ldots, n^d - 1$, where $d \in \{+, -\}$ is the evolution sequence to which the step belongs. The set $\Theta_k^d$ contains the tilting angles for the edges of the seed polygon $f_k^d$. This set must satisfy $|\Theta_k^d| = |V_k^d|$ (i.e., the number of tilting angles for the evolution step must be the same as the number of sides of the respective seed polygon). Additionally, two consecutive angles, $\theta_i, \theta_j \in \Theta_k^d$, must comply with $\text{sign}(\theta_i) = -\text{sign}(\theta_j)$ (i.e., tilting angles toggle their directions along the edges of the seed polygon). The toggling directions are equivalent to the edge directions required to generate interlocking blocks using the generation methods from Kanel-Belov et al. [3] and Bejarano and Hoffmann [7]. The scalar value $\lambda_k^d \in \mathbb{N}^+$ is the evolution range for the respective evolution step. Each $f_{k+1}^d = \text{EVOLVE}(f_k^d, \Theta_k^d, \lambda_k^d), \forall d \in \{+, -\}, \forall k = 0, 1, \ldots, n^d - 1$ defines the vertices and faces of the evolved polyhedron.

### 4.2. Fundamental TI Generation Requirement

Two neighboring polyhedra must have a common planar interface between them. Such an interface is the intersection of the respective faces from both polyhedra that are mutually coplanar. Let $f_i, f_j \in F$ be two neighboring tiles in a surface tessellation with shared edges represented as half-edges $ab, ba$ for the respective tile. The tilting angles $\theta_{ab}^+ \in \Theta_0^+$ and $\theta_{ba}^- \in \Theta_0^-$ (only for the first step from both evolution sequences) must be

$\theta_{ab}^{+} = -\theta_{ba}^{-}$. The tilting angles rotate the normal vector associated with the shared edge such that it defines the same tilted plane $P_{a,b}$ that contains the edge.

This requirement satisfies the cross-section criteria [2] for TI arrangements. For a planar tessellation composed of square tiles, each tile is the mid-section of the respective block in the assembly. A tile evolves into rectangles as it moves upward toward its normal vector. Eventually, the rectangles collapse into a line segment when the evolving tile reaches the topmost section of the block. Simultaneously, the tile also evolves into rectangles as it moves downwards in the opposite direction of its normal vector, and it will collapse into a line segment when the rectangles reach the bottom-most section of the block. The rectangles moving towards the top of the block are rotated $\frac{\pi}{2}$ angles concerning the rectangles moving towards the bottom of the block. Figure 13 shows an example of squares evolving into rectangles in an assembly of interlocked tetrahedra. Kanel-Belov et al. introduced the TI criterion using such an evolution principle [3].

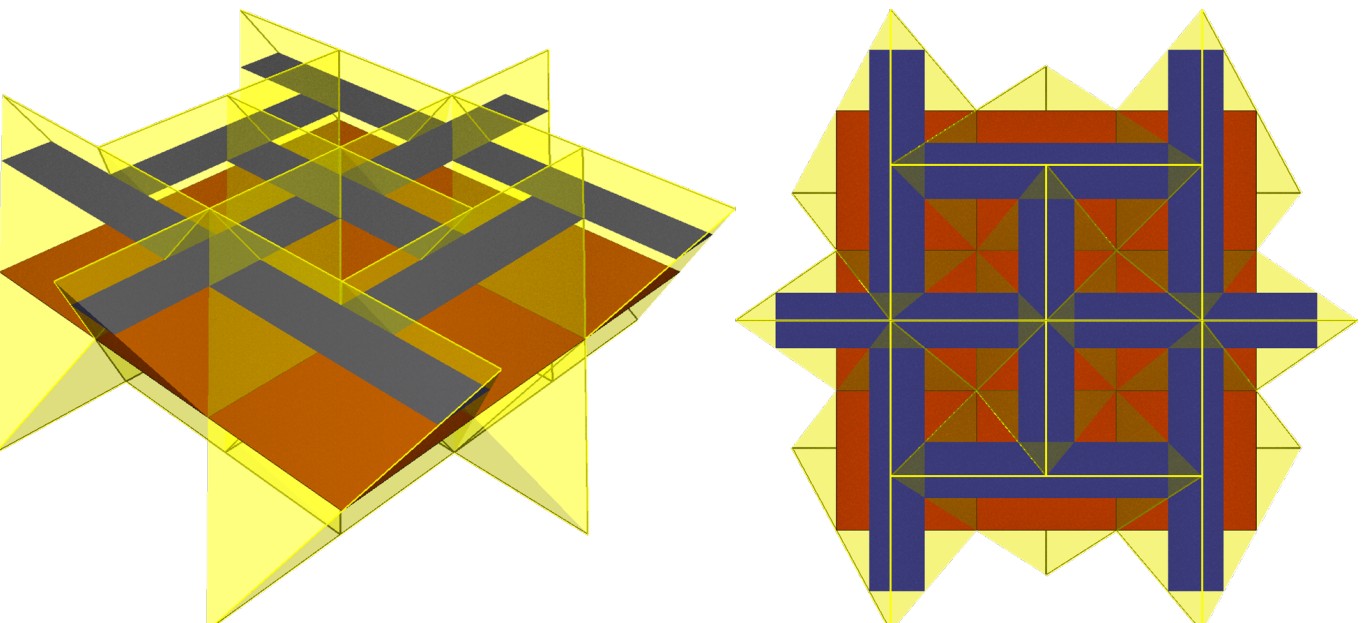

**Figure 13.** Squares evolving into rectangles while generating tetrahedra. (**Left**): assembly overview. (**Right**): top view showing the rectangles as cross-sections.

The polyhedron section contained between its first evolved $n$-polytopes, along with both positive and negative directions, contributes to the TI behavior of the resultant block. Such a section contains the faces in contact between the block and its neighbors. Its geometry is an antiprism without both top and bottom faces. This geometry guarantees the cross-section criteria of the block in the resultant assembly. The polyhedron sections defined after the first evolution steps (along with both positive and negative directions) can contribute to the TI behavior if their respective tilting angles are the same as those from the first evolution steps.

Figure 12a,c show two different polyhedra evolved from the same seed polygon. Both positive evolution sequences are identical. The negative evolution sequences differ in the last step, where the evolving rectangle collapses into a line segment. This change alters the common interface with a neighboring block in the assembly (if any). Furthermore, it breaks the uniformity of the block without sacrificing the TI property. Moreover, the volumes of the polyhedra become different (assuming uniform density). Changes in the evolution step parameters have implications for the distribution of forces required to maintain the static equilibrium of an assembly.

## 5. Results

This section describes generating different TI solids using the evolution-steps approach. We focus on the Platonic solids and insights to generate their truncated versions.

### 5.1. Platonic Solids

Dyskin et al. showed that all five Platonic solids have TI properties when placed in a way such that the tiles of the tessellation fit as the mid-sections of the solids [2]. Specifically, a square tile produces a tetrahedron. A hexagon tile produces either a hexahedron, an octahedron, or a dodecahedron. A decagon tile produces an icosahedron. If a tile is regular, the respective polyhedron can also be regular. Still, the tilting angles and Height–Bisection generation methods can only generate the tetrahedron and the octahedron. Generating the cube requires an additional step that intersects the planes tilted towards the same direction on the hexagon. No combination of intersecting planes generates all vertices for the dodecahedron and the icosahedron.

We can use uniform evolution to generate the Platonic solids, starting with a respective regular polygon. Each solid requires the same number of positive and negative evolution steps. The tetrahedron starts with a square and requires one step. The cube starts with a regular hexagon and requires two steps. The octahedron starts with a regular hexagon and requires one step. The dodecahedron starts with a regular decagon and requires two steps. The icosahedron starts with a regular decagon and requires two steps. The last steps for the cube, dodecahedron, and icosahedron must collapse the penultimate evolution polygon into a point. Figure 14 shows the positive evolution steps along $\hat{N}$ for each Platonic solid. Negative evolution steps are reciprocal. Table 2 lists the evolution parameter values required to generate each Platonic solid. The parameters are functions of the side length $l$ from the respective tile. The $\pm$ symbol indicates the sign of the angles alternating along the edges of the respective seed polygon.

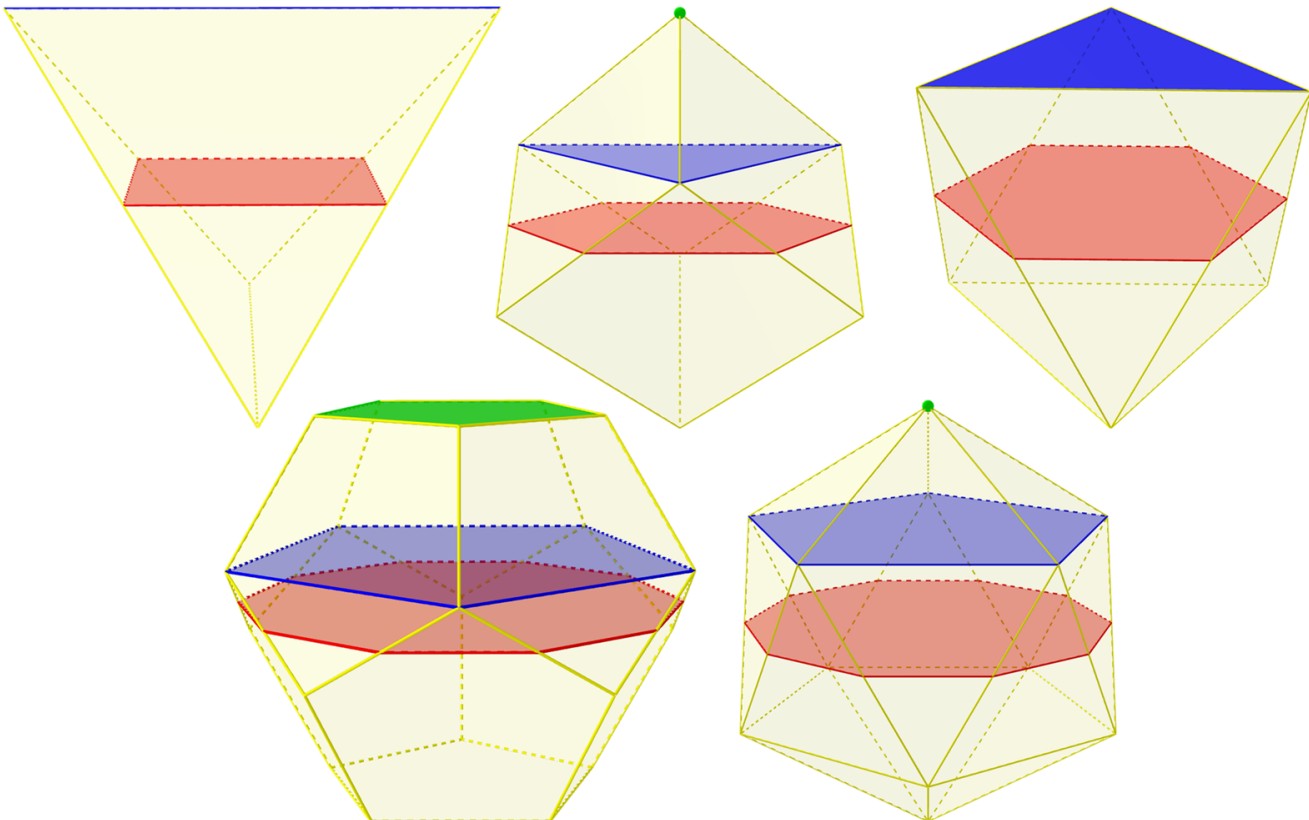

**Figure 14.** Positive evolution steps of the Platonic solids. Seed polygons are in red, evolved *n*-polytopes are in blue (from the first evolution step) and green (from the second evolution step).

**Table 2.** Evolution-step parameters to generate the Platonic solids. Seed polygons are regular and of side length $l$, and $d \in \{+, -\}$.

| Solid | Seed | Steps | Edge | Radius | $\Theta^d$ | $\lambda^d$ |
|-------|------|-------|------|--------|-----------|-------------|
| Tetrahedron | Square | 1 | $2l$ | $l\sqrt{\frac{3}{2}}$ | $\theta_0^d = \pm\left(\frac{\pi}{2} - \arctan\left(\frac{1}{\sqrt{2}}\right)\right)$ | $\lambda_0^d = \frac{l}{\sqrt{2}}$ |
| Cube | Hexagon | 2 | $l\sqrt{2}$ | $l\sqrt{\frac{3}{2}}$ | $\theta_0^d = \pm\left(\frac{\pi}{2} - \arctan\left(\frac{1}{\sqrt{2}}\right)\right)$ $\theta_1^d = \frac{\pi}{2} - \arctan\left(\frac{1}{\sqrt{2}}\right)$ | $\lambda_0^d = \frac{l}{\sqrt{6}}$ $\lambda_1^d = l\sqrt{\frac{2}{3}}$ |
| Octahedron | Hexagon | 1 | $2l$ | $l\sqrt{2}$ | $\theta_0^d = \pm\left(\frac{\pi}{2} - \arctan\left(\frac{1}{2\sqrt{2}}\right)\right)$ | $\lambda_0^d = l\sqrt{\frac{2}{3}}$ |
| Dodecahedron | Decagon | 2 | $\frac{2l}{\phi}$ | $l\sqrt{3}$ | $\theta_0^d = \pm\left(\frac{\pi}{2} - \arctan\left(\frac{1}{2}\right)\right)$ $\theta_1^d = \frac{\pi}{2} - \arctan\left(\frac{1}{2}\right))$ | $\lambda_0^d = l\sqrt{\frac{7-3\sqrt{5}}{5-\sqrt{5}}}$ $\lambda_1^d = l\sqrt{\frac{6+2\sqrt{5}}{5+2\sqrt{5}}}$ |
| Icosahedron | Decagon | 2 | $2l$ | $l\sqrt{\frac{5-\sqrt{5}}{3-\sqrt{5}}}$ | $\theta_0^d = \pm\left(\frac{\pi}{2} - \arctan\left(\frac{\sqrt{5}-2}{\sqrt{6-2\sqrt{5}}}\right)\right)$ $\theta_1^d = \frac{\pi}{2} - \arctan\left(\frac{1}{\sqrt{14-6\sqrt{5}}}\right)$ | $\lambda_0^d = l\sqrt{\frac{3-\sqrt{5}}{10-4\sqrt{5}}}$ $\lambda_1^d = 2l\sqrt{\frac{3-\sqrt{5}}{5-\sqrt{5}}}$ |

### 5.2. Truncated Platonic Solids and Clipped Solids

We can describe the truncated Platonic solids as multi-step evolved polygons as well. For example, the first two sub-figures from left to right in Figure 12 show a truncated tetrahedron generated using uniform evolution. We start with a square of side length $l$ to generate a regular truncated tetrahedron. The parameters for both positive and negative sections of the solid are reciprocal. The first-evolution-step parameters are $\Theta_0^d = \pm\left(\frac{\pi}{2} - \arctan\left(\frac{1}{\sqrt{2}}\right)\right)$ and $\lambda_0^d = \frac{l}{3\sqrt{2}}$. The second-evolution-step parameters are $\Theta_1^d = \frac{\pi}{2} - \arctan\left(\frac{1}{\sqrt{2}}\right)$ and $\lambda_1^d = \frac{2l}{3\sqrt{2}}$.

To generate the remaining truncated Platonic solids, we also use uniform evolution. The truncated cube requires three evolution steps. The truncated octahedron requires two evolution steps. The truncated dodecahedron requires three evolution steps. Finally, the truncated icosahedron requires four evolution steps.

Clipped solids also result from multi-step evolved polygons. Figure 7 shows how the evolution length parameter $\lambda$ limits the range of an evolution step. In such cases, we can stop an evolution step before the event of intersecting lines $L_i$ happens. Adjusting the $\lambda$ parameters is equivalent to clipping a block.

### 5.3. Mixed Solids

Different evolution step parameters for both positive and negative directions result in mixed geometries. From left to right, the last two subfigures in Figure 12 show a solid whose positive section resembles a truncated tetrahedron, while its negative section resembles a regular tetrahedron. Such a polyhedron still satisfies the TI requirements. To generate it, we start with a square of side length $l$. The positive evolution step parameters $\Psi_0^+$ are the same as the truncated tetrahedron, as mentioned in the previous subsection. The negative evolution step parameters $\Psi_0^-$ are the same as the tetrahedron, as listed in Table 2.

To illustrate the versatility and effectiveness of the proposed method, we constructed TICs using half-truncated tetrahedra (Figure 15) and truncated tetrahedra (Figure 16) based on various geometric domains. The considered domains are a planar grid, an elliptic paraboloid, a sinusoidal 3D wave, a saddle, a barrel vault, a sphere, and a torus. These examples were constructed using the Python Version 1.0 of TIGER [36] available at https://github.com/andresbejarano/tiger (accessed on 22 July 2024).

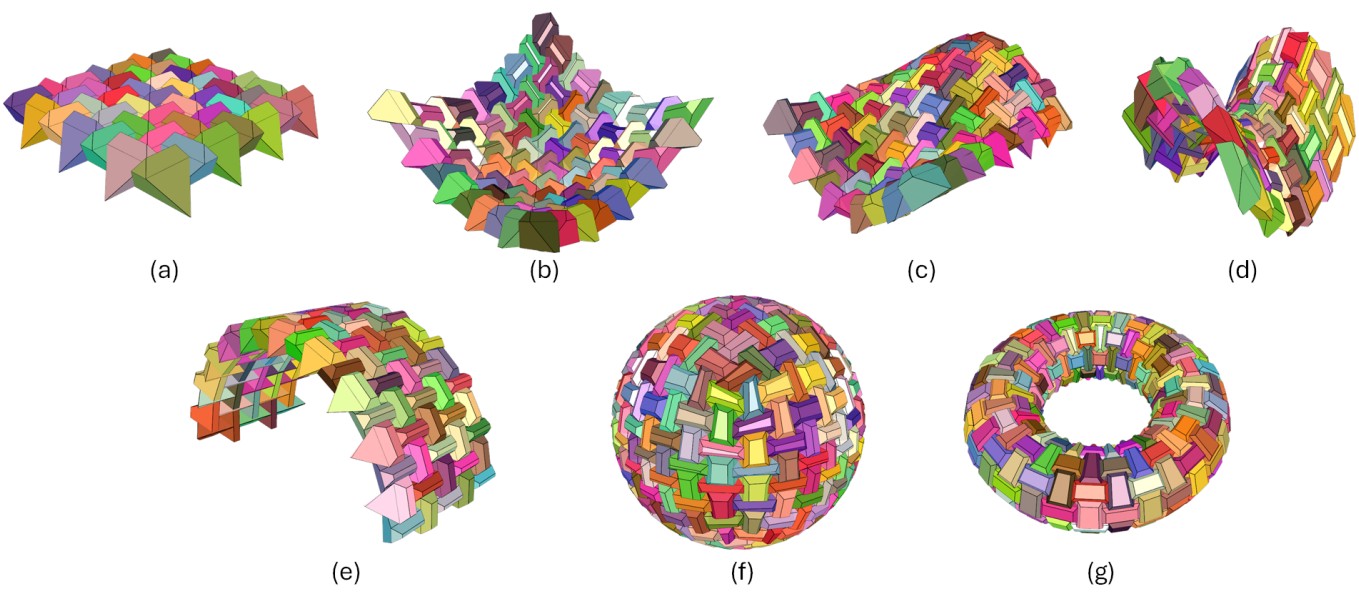

**Figure 15.** TICs built using half-truncated tetrahedra based on different geometric domains: (**a**) planar, (**b**) elliptic paraboloid, (**c**) wave, (**d**) saddle, (**e**) barrel vault, (**f**) sphere, and (**g**) torus.

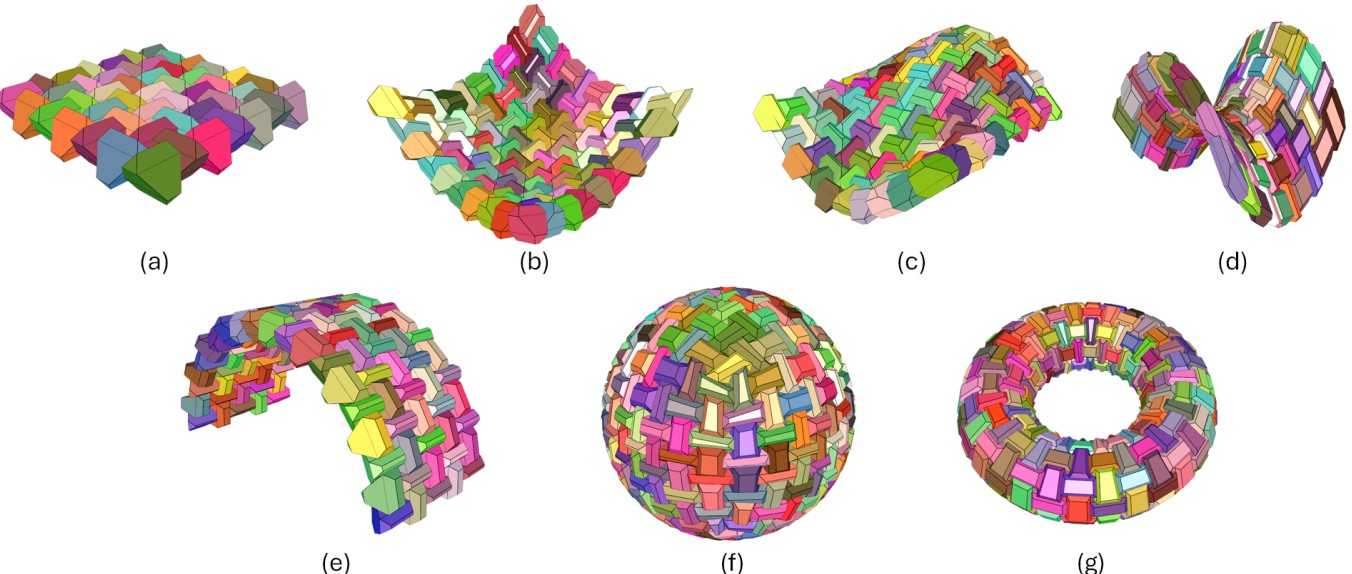

**Figure 16.** TICs built using truncated tetrahedra based on different geometric domains: (**a**) planar, (**b**) elliptic paraboloid, (**c**) wave, (**d**) saddle, (**e**) barrel vault, (**f**) sphere, and (**g**) torus.

*5.4. Concave Solids*

We can use the double-direction polygon–polyhedron evolution to generate concave solids with TI properties. As mentioned, the first positive and negative evolution steps must comply with the Fundamental TI generation requirement. An example of such concave solids was reported by Tessmann [23] and Tessmann and Becker [24]. Figure 17 shows the design from Philipp Mecke replicated using double-direction polygon–polyhedron evolution starting from an irregular hexagon. The design of such blocks involves subtracting sections from a tetrahedron while maintaining the interlocking principle. The parametric design of Mecke's block allows for the generation of TI assemblies with adjustable porosity (i.e., holes between adjacent blocks).

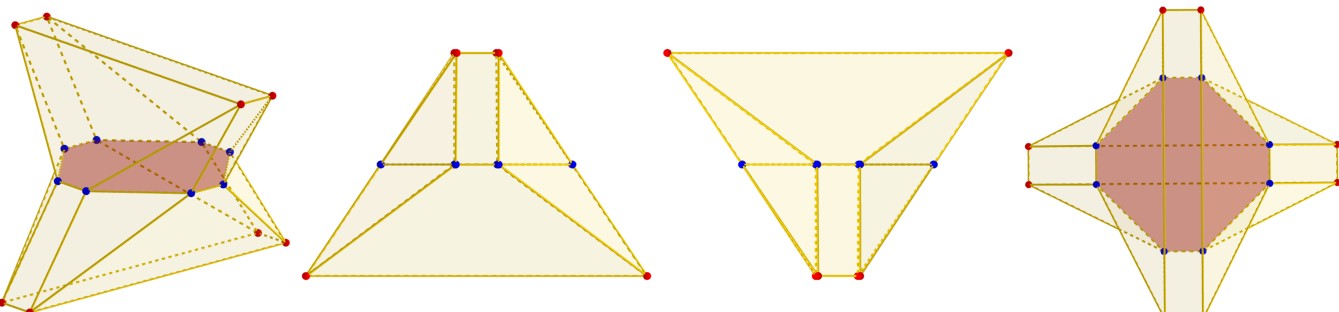

**Figure 17.** TI polyhedron, by Mecke, generated using double-direction polygon–polyhedron evolution. From **left** to **right**: oblique view, front view, right view, and top view.

To generate Mecke's TI block, we start with an irregular hexagon of side lengths $a$ (shorter side length) and $b$ (longer side length) as the seed polygon $f$. To keep our approach as parametric as possible, we set the vertices of $f$ to lie in a circumference of radius $r$. Side length $a$ is a parameter, which leaves $b$ as a function of $r$ and $a$. We only required one evolution step along the positive and negative sides of $f$ to generate the block. We defined angles $\theta_a$ and $\theta_b$ for their respective side lengths. All angles had to be positive, but two were respective to a pair of opposite short side lengths. The other pair of opposite short side lengths had to be negative for the negative evolution step. Infinitely many parameter values exist to generate the block using double-direction polygon–polyhedron evolution. The result shown in Figure 17 has values of $r = 1$, $a = 0.4$, $b = 1.11$, $\theta_a = 0.98$ radians, and $\theta_b = 1.13$ radians. Evolution length $\lambda$ had to be long enough to allow the lines $L_i$ to intersect within the evolution range.

Tessmann and Becker continued exploring Mecke's TI polyhedron by considering its performance and gradually reducing its mass [24]. We replicated the equivalent geometries even when no mass could be removed. We used the same evolution parameters as the original shape for such a block shape. Still, we reduced the length of the shorter irregular hexahedron edges until the resultant polygon was a square. Figure 18 shows the mass-decreased Mecke's block replicated using double-direction polygon–polyhedron evolution.

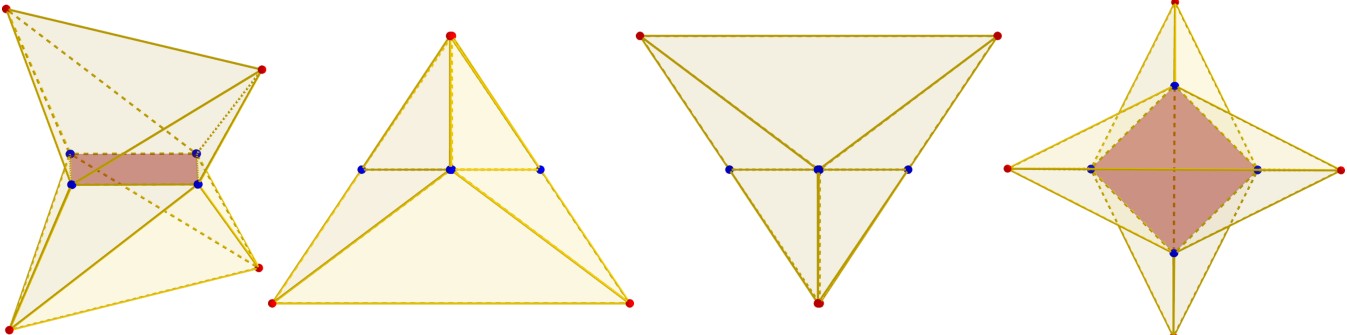

**Figure 18.** Collapsed polyhedron, by Mecke, generated using double-direction polygon–polyhedron evolution. From **left** to **right**: oblique view, front view, right view, and top view.

## 6. Conclusions

This article has proposed a framework to generate TI polyhedra as a multi-step evolution process. The framework generalizes over the tilting angles and Height–Bisection generation methods. An evolution step evolves (i.e., reshapes and translates) a seed polygon into a $n$-polytope (e.g., point, line segment, or polygon). Consecutive evolution steps are possible when the evolved $n$-polytope is a polygon. Possible evolution steps could follow a single or double direction to the normal vector of the seed polygon. Additionally, reciprocal and uniform evolution steps produce different shapes, which comply with the TI requirements under certain matching specifications. The TI shapes generated with the

framework include the Platonic solids (and truncated versions), mixed solids, and concave TI blocks.

The parametric nature of the framework opens the door for further discussions about TI assemblies and their functionality. For example, how to increase block volumes to redistribute compression/tension forces on the assembly and expand the catalog of possible interlocking configurations that comply with the topological interlocking principle. Our proposed method adapts to various surface requirements, allowing for a high degree of customization and versatility in design. Engineers and architects can leverage this flexibility to create intricate and aesthetically pleasing structures that meet specific functional and architectural needs. This ability to generate unique, interlocking designs that are both functional and visually appealing can benefit applications in custom architecture, modular construction, and artistic installations.

The Evolution Steps method can enhance the design and manufacture of interlocking assemblies in mechanical and materials engineering. Architects, designers, and engineers can create customized, interlocking structures that exhibit enhanced mechanical properties by enabling the precise control of block shapes through polygon transformations. Such mechanical properties include an increased load-bearing capacity, improved energy absorption, and superior resistance to deformation. These characteristics are advantageous in applications requiring robust and resilient materials, such as protective gear, impact-resistant panels, and aerospace components.

One critical advantage of the proposed Evolution Steps method is its potential to minimize material waste during the manufacturing process. Traditional construction techniques often involve cutting and shaping materials, leading to significant waste. In contrast, our method allows for the precise design of interlocking blocks that fit with minimal gaps, reducing the need for excessive cutting and trimming. This efficiency conserves resources and lowers production costs and environmental impact, aligning with sustainable manufacturing practices.

Additional research is needed to expand our understanding of the TI generation process in terms of physical requirements, for example, to find the optimal evolution scheme and parameters to minimize tension forces between blocks for the assembly to reach static equilibrium.

**Author Contributions:** Conceptualization, A.B. and K.M.; methodology, A.B.; software, A.B. and K.M.; validation, A.B.; formal analysis, A.B.; investigation, A.B. and K.M.; resources, A.B. and K.M.; data curation, A.B.; writing—original draft preparation, A.B.; writing—review and editing, A.B.; visualization, A.B. and K.M.; supervision, A.B.; project administration, A.B. All authors have read and agreed to the published version of the manuscript.

**Funding:** This research received no external funding.

**Institutional Review Board Statement:** Not applicable.

**Informed Consent Statement:** Not applicable.

**Data Availability Statement:** The original contributions presented in the study are included in the article, further inquiries can be directed to the corresponding author.

**Conflicts of Interest:** The authors declare no conflicts of interest.

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
