# Peer review of "Multistep Evolution Method to Generate Topological Interlocking Assemblies"

_applsci, doi:10.3390/app14156542_

Round 1

Reviewer 1 Report

Comments and Suggestions for Authors

I found this paper interesting and well organised, but a bit too formal. I suggest to consider the following comments to improve the quality of the paper.

-State of the art: please provide references to related research with practical use of interlocking assembly such as: 10.1016/j.autcon.2020.103117, 10.1016/j.istruc.2024.106156, 10.1145/3592095

-Please improve the figures (in particular fig. 13), since they are very ambiguous. It is not clear what is in front and back.

-In section 5 add some examples of 3D applications, e.g., plates and shell that are tiled with selected patterns.

-Please doublecheck the numbers of figures and sections when referenced in the manuscript.

Comments on the Quality of English Language

ok

Author Response

We appreciate your comments and time to review our proposed paper.

Comments 1: State of the art: please provide references to related research with practical use of interlocking assembly such as: 10.1016/j.autcon.2020.103117, 10.1016/j.istruc.2024.106156, 10.1145/3592095
Response 1: The mentioned papers, plus additional recent papers on the generation of TICs from a geomtric approach, were added to the document.

Comments 2: Please improve the figures (in particular fig. 13), since they are very ambiguous. It is not clear what is in front and back.
Response 2: Figure 13 has been rendered again with a different graphics engine.

Comments 3: In section 5 add some examples of 3D applications, e.g., plates and shell that are tiled with selected patterns.
Response 3: We have included additional examples of shells and plates generated using the Evolution Steps to create different block shapes with TIC behavior.

Comment 4: Please doublecheck the numbers of figures and sections when referenced in the manuscript.
Response 4: Numbers of figures and sections have been revised and fixed accordingly.

Reviewer 2 Report

Comments and Suggestions for Authors

The manuscript introduces an innovative framework for generating blocks in Topological Interlocking (TI) assemblies. The authors present a generalized method that enhances previous TI generation techniques through a multistep evolution process. This method involves translating and reshaping a seed polygon through evolution steps, resulting in various TI blocks encompassing new and previously known shapes. The manuscript addresses a crucial and underexplored aspect of topological interlocking by providing a comprehensive approach to generating TI blocks from fundamental geometric principles. The proposed framework is promising for theoretical and practical structural engineering and computational design advancements.

The manuscript represents a notable and meaningful contribution to topological interlocking. It is suitable for publication with minor revisions.

Suggestions for Revision

1. Enhance the clarity of figures and provide more detailed captions.

2. Elaborate on potential real-world applications of the proposed method.

3. Include a discussion on the computational complexity and scalability of the method.

4. Proofread the manuscript to correct minor typographical errors and improve overall readability.

5. The references are old and the latest ones need to be cited.

Author Response

We appreciate your comments and time to review our proposed paper.

Comments 1: Enhance the clarity of figures and provide more detailed captions.
Response 1: We have increased the clarity of critical figures, including Figure 13 (which was ambiguous due to rendering artifacts)

Comments 2: Elaborate on potential real-world applications of the proposed method.
Response 2: We have included visual examples of TICs created with the Evolution Steps method based on different surfaces. Also, we discuss potential applications of our proposed method in the Conclusions section.

Comments 3: Include a discussion on the computational complexity and scalability of the method.
Response 3: We have included the Evolve algorithm in pseudocode and described its runtime complexity.

Comments 4: Proofread the manuscript to correct minor typographical errors and improve overall readability.
Response 4: Document has been proofread. We fixed typos and increased the readibility of critical statements of the document.